# Maneuverability of the Scope and Instruments within Three Different Single-Incision Laparoscopic Ports: An Experimental Pilot Study

**DOI:** 10.3390/ani11051242

**Published:** 2021-04-26

**Authors:** Georg Haider, Ursula Schulz, Nikola Katic, Christian Peham, Gilles Dupré

**Affiliations:** 1Small Animal Clinics, Vetmeduni Vienna, 1210 Vienna, Austria; gillespierre.dupre@gmail.com; 2Large Animal Clinics, Vetmeduni Vienna, 1210 Vienna, Austria; ursula.schulz@vetmeduni.ac.at (U.S.); christian.peham@vetmeduni.ac.at (C.P.); 3Vet Chirurgie, 1210 Vienna, Austria; Nikola.katic@vetchirurgie.at

**Keywords:** maneuverability, range of motion, single-port access system, SILS, glove port

## Abstract

**Simple Summary:**

Single-port access systems, used to perform endoscopic surgery through a single incision, are currently used in many human and veterinary surgeries. These systems present some technical challenges as they offer limited space for manipulation. No objective study has evaluated the degree of possible movement within different single-port access systems. This study aimed to measure and compare the possible movements of the endoscope and instruments within three single-port access systems: the Covidien SILS-port, Storz Endocone, and glove port. The glove port consists of a surgical glove and an O-ring retractor. A clear acrylic box with artificial skin was used to mimic the abdominal wall and cavity. A motion capture system with 18 cameras was used to trace the possible movements of the endoscope. The volume of the cone-shaped three-dimensional figure described by the scope when circled 360° was used to compare maneuverability across the three systems. The glove port showed higher maneuverability than the two commercially available ports when an endoscope alone, or an endoscope and instrument, was or were inserted. A higher degree of maneuverability is positively associated with easier handling of the endoscope and instruments during surgical procedures. The findings of this study may benefit students and young professionals by helping them to select an easy-to-use single-port access system when beginning single-incision endoscopic surgery.

**Abstract:**

Single-port access systems (SPASs) are currently used in human and veterinary surgeries. However, they pose technical challenges, such as instrument crowding, intra- and extracorporeal instrument collision, and reduced maneuverability. Studies comparing the maneuverability of the scopes and instruments in different SPASs are lacking. This study aimed to compare the maneuverability of three different SPASs: the Covidien SILS-port, Storz Endocone, and glove port. A clear acrylic box with artificial skin placed at the bottom was used to mimic the abdominal wall and cavity. The three SPASs were placed from below, and a 10-mm endoscope and 5-mm instrument were introduced. A motion analysis system consisting of 18 cameras and motion analysis software were used to track the movement of the endoscope and instrument, to determine the volume of the cone-shaped, three-dimensional figures over which movement was possible, with higher values indicating greater maneuverability. The Mann–Whitney U test was used for the analysis. The maneuverability of the endoscope alone was significantly higher in the glove port system than in the other two SPASs. When inserting an additional instrument, the maneuverability significantly decreased in the SILS-port and Endocone, but not in the glove port. The highest maneuverability overall was found in the glove port.

## 1. Introduction

Laparoscopic surgery improves the visualization of intraabdominal organs, reduces infection rates and postoperative pain, and reduces the postoperative recovery period in animals [1,2,3]. In human surgery, single-port access systems (SPASs) were developed to reduce the number of incisions while maintaining an adequate number of cannulas to perform the surgery [4]. In recent veterinary studies, the use of SPASs has been described in various surgical procedures, such as ovariectomy, ovariohysterectomy, combined ovariectomy/gastropexy, and cryptorchidectomy [5,6,7] However, the loss of triangulation, instrument crowding, and clashing are inherent disadvantages of SPASs, which increase the difficulty of carrying out certain procedures and result in longer surgeries [4]. An additional disadvantage of the commercially available SPASs is the relatively high cost.

A cost-effective alternative to commercial SPASs was described using an adjustable O-ring retractor and a surgical glove [8]. Recently, this technique was successfully used to conduct an ovariohysterectomy in a dog treated for pyometra [9,10]. Furthermore, the duration of surgery using a homemade glove port was shorter than other single-incision laparoscopy techniques, or multi-port laparoscopy, when used on normal uteri [10]. Another study confirmed these findings by showing that elective procedures are faster when single-incision laparoscopic surgery (SILS) ports (Covidien, MA, USA) are used. Furthermore, this technique results in fewer intra-operative complications when performed by surgeons experienced in laparoscopy [11].

Although the learning curve for laparoendoscopic single-site (LESS) surgeries has been described as short and defined for experienced veterinary surgeons using a SILS-port [11], surgical novices performed better using SPASs other than the SILS-port, such as the GelPoint Access System [12]. The subjectively higher maneuverability was likely achieved by the use of different systems for abdominal wall retraction [12]. Similar to the GelPoint Access System, the glove port uses an adjustable O-ring retraction system for abdominal retraction, whereas the SILS-port passively conforms to the abdominal wall without active retraction. In addition, instrument circling, described as the ability to move the endoscope or instrument around within the SPAS, is possible in the glove port and GelPoint access port, whereas instrument circling is impossible in the SILS-port. This study aimed to objectively compare the possible movements of the endoscope and endoscopic instruments using three different SPASs: the Covidien SILS-port, Storz Endocone, and glove port.

## 2. Materials and Methods

The SILS-port is a commercially available flexible soft-foam port that maintains pneumoperitoneum by conforming to the patient’s abdominal wall. The bottom half of the port is inserted through a 40-mm incision using an atraumatic clamp. The port can either hold three 5-mm cannulas or one 12- or 15-mm cannula and two 5-mm cannulas. The port is removed by simply pulling upward.

The glove port is an improvised single-incision port consisting of an O-ring abdominal retractor (Alexis O Wound Retractor; Applied Medical, Rancho Santa Margarita, CA, USA) and a surgical glove. The port is assembled as previously described [10]. To achieve comparable results, the same 5-mm SILS cannulas were used to assemble the glove port. The O-ring retractor was placed through a 40-mm incision into the artificial skin, and the glove and cannulas were secured on the external ring.

The Karl Storz Endocone (Karl Storz, Tuttlingen, Germany) consists of a cone-shaped stainless-steel cylinder and a removable multiport-plate closing the cone. This multiport-plate consists of three 15-mm and six 5-mm ports. The distal part of the cylinder dwells on a single large thread to facilitate easy abdominal insertion. As with the two other ports, the distal part of the Endocone was introduced through a 40-mm artificial abdominal wall incision.

The experimental setup consisted of a custom-made clear acrylic box (1 × 1 × 0.5 m) with an artificial abdominal wall (Limbs & Things, Bristol, UK) centered at the bottom of the box so as not to obstruct the ceiling-mounted cameras from viewing the instruments. To measure the maneuverability of the scope and the instruments, the clear acrylic box was centered in the middle of a lab environment with an 18-camera motion capture system (10 Eagles, 8 Kestrel 300, 60 Hz, Cortex V7.0, Motion Analysis Corporation, Rohnert Park, CA USA) that recorded the movements of the endoscope and instruments (Figure 1).

Each single-port device was introduced from below, according to the manufacturer’s guidelines. A 10-mm dummy endoscope (10-mm round, 100-cm long wooden dowel) was inserted into the respective port. The maximal working length was set to 31 cm, which matches the working length of a 10-mm, 0° Storz endoscope. Three spherical reflective markers were mounted on the dummy endoscope to track the movement of the scope. One marker was mounted on each end, and the third marker was mounted 15–20 cm from the inner tip of the dummy endoscope or instrument (Figure 1 only shows the two inner markers).

To define the origin in three-dimensional space, reference points (spherical reflective markers) were placed on four corners of the clear acrylic box. These corners were selected at the beginning of the experiment and were unaltered throughout the experiment. The maximal depth of penetration, possible angle, and possible movement of the scope were measured by tracking the movement of the spherical reflective markers with the motion capture system. Once measurements had been recorded with an endoscope only, the endoscope was fixed in a vertical position, and a 5-mm dummy instrument was inserted into the working channel of the respective port. Like the endoscope, the instrument was equipped with three spherical reflective markers, and its maneuverability was recorded while the endoscope remained in the fixed vertical position. The instrument was moved as much as possible without moving the respective SPAS within the artificial skin, and these movements were recorded using the motion capture system. Each measurement was repeated six times on two different days.

The recorded data were processed using motion capture software (Cortex V7.0, Motion Capture Analysis, Rohnert Park, CA, USA) and thereafter transferred to MATLAB R2015a (MathWorks, Natick, MA, USA). Using MATLAB, the software was programmed to calculate the volume, maximum depth, and angles using the data measured by the motion capture system. The maximal depth of penetration was calculated as the distance between the top marker of the endoscope or instrument and the level of the artificial skin, indicated by the marker placed at the bottom of the clear acrylic box.

The maneuverability was set as the calculated volume (volume of maneuverability, VoM) covered by the maximal possible movement of the endoscope or the instrument. A three-step calculation method was used. First, a stereolithography (.stl) model, describing the surface of the volume covered as coordinates, was generated. To achieve this, the trajectories of the two inner markers of the endoscope or the instrument at the bottom level of the box were used. This resulted in a volume composed of a cone topped with a half sphere. In the second step, the stereolithography model was converted into a polygon file format using the “Poisson-dis-sampling” method in MESHLAB (Version 1.3.3 MeshLab, meshlab.net Accessed on 30 April 2018). In the third step, the polygon file was imported to AMIRA (Version 5.3.3, Thermo Fisher Scientific, Waltham, MA, USA) to generate the body of a polygon, and the volume was calculated. To reduce possible errors, the volumes were also computed with MESHLAB and compared with the results from AMIRA. Appendix A can be found in Appendix A.

The area of maneuverability (AoM) was used as a second indicator of maneuverability and was defined as the area described by the tip of the endoscope or instrument during a 360° movement within the respective port. This area was equivalent to the surface area of the half sphere of the resulting VoM used in the previous three steps. The AoM was measured in MESHLAB by selecting all triangles of the half cone in the stereolithography file (Figure 2). Appendix A can be found in Appendix A.

The maximal angles were measured in MESHLAB using the cone-shaped volumes obtained from the stereolithography files. The maximal angle was defined as the angle between the outer borders of the cone (equivalent to the trajectory of the endoscope) and was measured in the projection of the cone to the X- and Y-plane. Therefore, the maximal angle measured at the X-plane was equivalent to the sum of the angles at 0° and 180°, and the maximal angle measured at the Y-plane was equivalent to the sum of the angles at 90° and 270°. Appendix A can be found in Appendix A.

Statistical outliers were defined as values that differed by >5% between two measurements, or values above twice the mean or below half of the mean. These were removed from the dataset. The remaining values were analyzed using the Mann–Whitney U test. The level of significance was set to 5% (*p* = 0.05).

## 3. Results

The median depth of penetration was highest with the Endocone (323 mm; range: 318–323), followed by the SILS-port (288 mm) and then the glove port (272 mm) (*p* < 0.05 between all variables). The median VoM with the endoscope alone differed significantly between all ports. The glove port VoM value was 18,199 cm^3^ (range: 14,184–20,136) compared with 9999 cm^3^ (range: 8574–10,937) and 1890 cm^3^ (range: 1665–2104) when using the SILS-port or Endocone, respectively (Figure 3). The largest AoM with an endoscope alone was observed when using the glove port (2017 cm^2^), followed by the SILS-port (1206 cm^2^) and the Endocone (221 cm^2^). The highest angles with an endoscope alone were also obtained within the glove port, followed by the SILS-port and the Endocone (Table 1).

After introducing the dummy endoscope, the VoM decreased significantly in all ports except the glove port. The reduction of VoM was the highest for the single-incision laparoscopic surgery (SILS)-port (1.98-fold), followed by the Endocone (0.75-fold). In contrast, the VoM increased while using the glove port. The VoM was 30,188 cm^3^ (range: 19,696–30,754) for the glove port, 8878 cm^3^ (range: 7361–10,622) for the SILS-port, and 850 cm^3^ (range: 623–947) for the Endocone (Figure 4).

Interestingly, the maximal angles increased significantly in the X- and Y-planes when using the glove port, and in the Y-plane after inserting an additional instrument when using the SILS-port (*p* < 0.05). In contrast, the maximal angle measured within the Endocone decreased significantly in the X-plane (Table 2).

## 4. Discussion

In this study, the improvised glove port showed the highest maneuverability (VoM and AoM) with an endoscope alone and with an endoscope with an instrument when compared with the Endocone and SILS-port. This finding was likely caused by differences in the construction of the ports. First, the difference in abdominal wall retraction will influence the area of the abdominal opening. While the SILS-port used flexible soft-foam, the glove port used an O-ring retractor, and the Endocone used a metal cone for abdominal wall retraction. The round and rigid shape of the Endocone enabled a round, standardized abdominal retraction. In contrast, the soft construction of the glove port and the SILS-port resulted in an ellipse-shaped abdominal wall retraction [13]. Second, the type of fixation of the cannulas within the port can prevent, or on the contrary, facilitate instrument circling within the SPAS. The SILS-port holds the instrument and the endoscope cannulas at the level of the abdominal wall, forcing the instrument to circle within this port. In contrast, the Endocone and the glove hold the instrument and the endoscope extra-abdominally, allowing instrument circling at the level of the abdominal wall. However, the rigid platform for the instruments and the scope in the Endocone negates this advantage, whereas the flexibility of the fingers in the glove port increases the effect of instrument circling, describing the ability to move an instrument and endoscope around each other within a SPAS at the level of the abdominal wall. This unique combination likely results in the higher instrument and endoscope maneuverability measured with the glove port.

When inserting both an endoscope and an instrument, the VoM decreased with both the commercially available SPASs (Endocone and SILS-port), whereas the AoM only significantly decreased with the Endocone. This was likely the result of the internal and external instrument clashing seen with these two SPASs (Figure 4a,b). In contrast, maneuverability increased in the glove port when an instrument and an endoscope were inserted. Theoretically, an increase in the volume of a cone can result from two different parameter changes: first, the height of the cone representing the depth of penetration in our model, and second, the radius of the cone representing the maximal possible angle in our model. Interestingly, when comparing the maximal angles, an increase could be seen in the glove port and the SILS-port after an endoscope and an instrument were simultaneously inserted, whereas both the angles decreased with the Endocone. This could be the result of unintended movement of the artificial skin during the measurement when both the instruments and endoscopes were inserted. Due to the flexible accommodation of the cannulas holding the instrument and the endoscope when using the glove port, a deeper penetration during the measurements was observed. The combination of these two changes during the measurements likely resulted in increased maneuverability when both the instrument and the endoscope were set in the glove port.

To the best of our knowledge, this is the first study comparing the maneuverability of an endoscope and instruments within three different SPASs. Our results demonstrate higher maneuverability in the glove port than that of the other two SPASs tested. A previous study reported better surgical performance for SPASs using an O-ring retractor for abdominal wall retraction owing to its subjectively higher instrument maneuverability than other SPASs [12].

There are some limitations to this study owing to the study design. First, only three SPASs were evaluated. Whether other commercially available or improvised SPASs provide higher maneuverability than the ports tested is unknown. Second, the experimental setup was quite different from an actual surgical setup. Unfortunately, the use of this motion capture system dictated these differences. The clear acrylic box had to be placed upside down so as not to obstruct the cameras from viewing the instruments, and we had to use wooden dummy instruments and endoscopes so as not to influence the motion capture system through reflections from the metal instruments. Additionally, the wooden dummy instruments and endoscopes differ in their external shape from those used in surgery. For example, instrument handles and camera heads were not simulated, and those likely increase external instrument clashing and reduce the maneuverability. Even if we used a marker to restrict the working length of the instrument and endoscope, this might not reflect the restriction of the working length in a true surgical setup. Another limitation is the selection of the size of the endoscope and the number of instruments. Since it is impossible to simultaneously measure the movement of two instruments, we restricted our experimental setup to one instrument. Therefore, we used a 10-mm dummy endoscope with a 5-mm dummy instrument to mimic the use of a 10-mm endoscope with a 6-mm working channel, as this setup, in combination with a glove port, was often used for laparoscopic-assisted gastropexy in our institution. The use of a 5-mm endoscope instead of a 10-mm endoscope in this experimental setup might increase the AoM and VoM within the SILS-port and Endocone, but is unlikely to increase the AoM and VoM of the glove port. Further studies comparing the differences in internal and external instrument clashing when using real instruments, as well as comparing different SPAS during standard endoscopic training tasks, are required. Third, articulated instruments are designed for use within different SPASs and could increase the degree of freedom [14]. In this study, articulated instruments were not investigated; thus, it is unknown if the use of articulated instruments could have influenced the study results.

## 5. Conclusions

The results of this study show that the improvised glove port allows higher maneuverability (VoM and AoM) than the Storz Endocone and Covidien SILS-port. This is likely the result of improved instrument circling and greater maneuverability of the cannulas.

## Figures and Tables

**Figure 1 animals-11-01242-f001:**
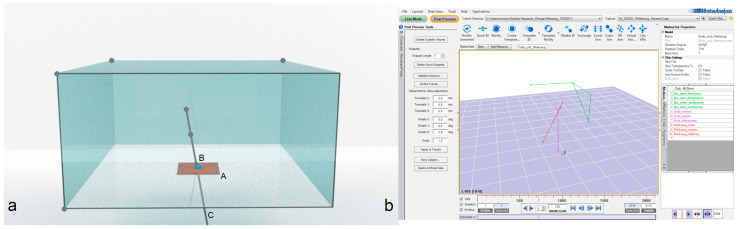
(**a**) Schematic drawing of the experimental setup. The artificial skin (A) holding the different ports (B) was mounted at the bottom of a clear acrylic box. A 10-mm endoscope (C) was inserted, and the maximal possible movement was recorded by 18 motion capture cameras. The positions of the reflective markers are shown as grey dots. (**b**) A picture of the recorded data recorded by the motion capture cameras.

**Figure 2 animals-11-01242-f002:**
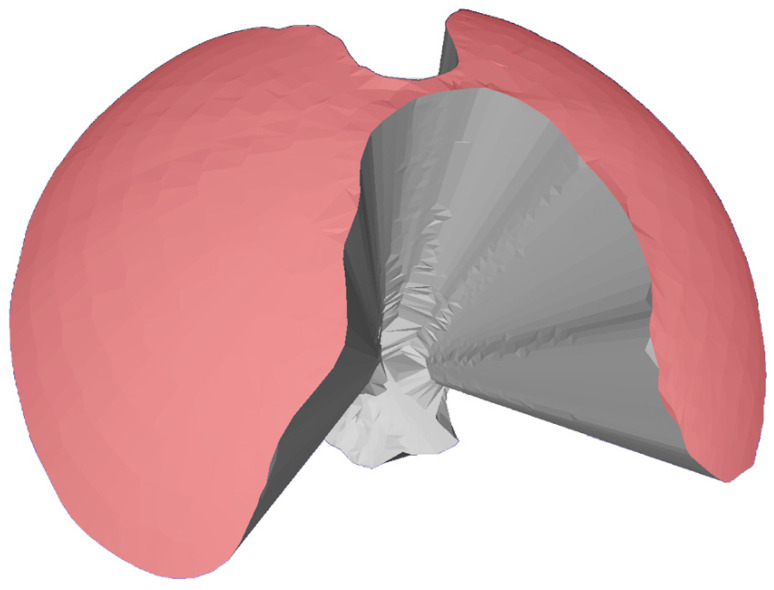
A rendered image from MESHLAB was used to manually select the upper area (red) to calculate the area of maneuverability. The example shows the measured volume of one SILS-port with both the endoscope and the instrument inserted. The image was rendered in Blender (www.blender.org, accessed on 30 April 2018).

**Figure 3 animals-11-01242-f003:**
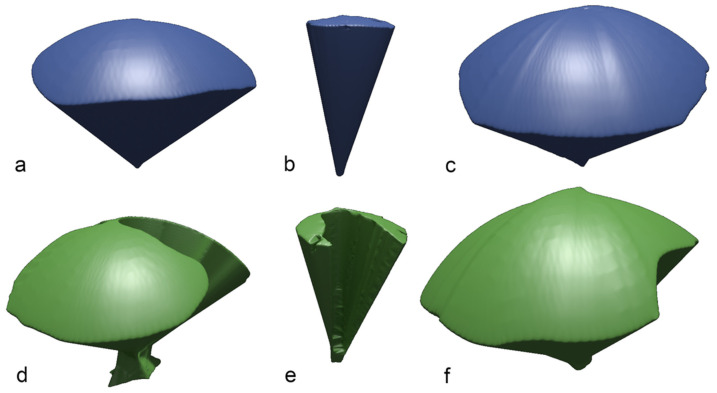
Example of rendered images of the volume of maneuverability (VoM). The upper line shows the VoM when only an endoscope was inserted into the ports. The lower row shows VoM when an endoscope and an instrument are inserted into the ports. (**a**,**d**) SILS-port; (**b**,**e**) Endocone; (**c**,**f**) glove port. The images were rendered in Blender (www.blender.org, accessed on 30 April 2018).

**Figure 4 animals-11-01242-f004:**
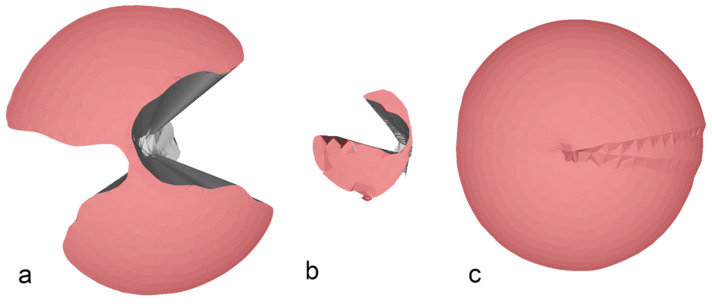
Examples of the area of maneuverability of the SILS-port (**a**), the Endocone (**b**), and the glove port (**c**) when an endoscope and an instrument were inserted in the ports. Note the reduced area of maneuverability caused by instrument interference in the SILS-port and the Endocone, but not in the glove port (images rendered with Blender).

**Table 1 animals-11-01242-t001:** Maximal measured angles in the X- and Y-plane within the respective ports when a 100-mm endoscope was inserted.

Maximal Angles	Storz Endocone		Covidien SILS-Port		Glove Port	
	X-Plane	Y-Plane	X-Plane	Y-Plane	X-Plane	Y-Plane
Median	36°	33°	94°	96°	114°	119°
Min	32°	31°	85°	87°	103°	98°
Max	39°	36°	104°	102°	124°	124°

SILS—the single-incision laparoscopic surgery.

**Table 2 animals-11-01242-t002:** Maximal measured angles in the X- and Y-planes within the respective ports when an endoscope (10 mm) and instrument (5 mm) were inserted.

Maximal Angles	Storz Endocone		Covidien SILS-Port		Glove Port	
	X-Plane	Y-Plane	X-Plane	Y-Plane	X-Plane	Y-Plane
Median	27°	31°	108°	119°	125°	125°
Min	25°	28°	90°	104°	107°	114°
Max	31°	38°	124°	129°	130°	129°

## Data Availability

Data available in a publicly accessible repository.

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
