# Peer review of "Maneuverability of the Scope and Instruments within Three Different Single-Incision Laparoscopic Ports: An Experimental Pilot Study"

_animals, 2021, doi:10.3390/ani11051242_

Round 1
Reviewer 1 Report
My overall recommendation is somewhere between serious revisions needed and rejection.
REVIEW AND COMMENTS
When deciding to review this article I thought I would be able to read the article and make a few constructive comments with a recommendation to publish. I am sorry but I am having great difficulty following course.
I am very confused by the design of this study. Placing the skin at the bottom of the chamber does not make any sense to me. This is upside down. In reality the skin is at the top, not at the bottom. The important instrument movement is within the abdomen and not outside of the abdomen. Fake instruments were used, and this could significantly affect instrument movement ability. The camera head on the telescope and instrument handles have an affect on instrument movement. I know that one or all of the authors have access to real instruments and not using the real thing makes no sense. A ten-millimeter diameter dowel was used for a telescope example but 5mm telescopes are the most common size laparoscope used in small animal practice. A smaller telescope could significantly increase mobility within the abdomen in the two commercially available single port systems. A telescope and a single instrument example was used whereas almost all minimally invasive surgeries require at least a telescope and two instruments. This would substantially affect instrument mobility and the single instrument example here does not measure reality.
One of the systems mentioned is the GelPoint Access System but this was not included in the study. Why not? Since this system was not studied it should not be mentioned.
I know that we as veterinarians are always looking for a cheaper alternative. But with the ability to reuse the commercially available systems the cost is not that great, and the use of a jury-rigged alternative may or may not be a good idea. We are not that poor.
The single portal minimally invasive surgery systems are intended to be used with specially designed curved instruments and we are trying to use them with traditional straight instruments. This may be a more valid reason for circumventing the intended design but is still an unfair categorization of the effectiveness of commercially available systems.
An unimportant but context error is that Acrylic is not glass. Acrylic is plastic. A more accurate term would be clear Acrylic.
Another issue is that the mobility needed to perform minimally invasive surgery has not been determined. This is a question that needs to be answered as part of this discussion.
The depth of insertion measurements are greater than the length of commonly used telescopes so how are these measurements relevant.
I also admit that I am biased against the single port idea for minimally invasive surgery. I have been performing minimally invasive surgery for a long time and have found no problems with the traditional multiport system. One of the advantages of the multiport system is that none of the incisions are large enough for dehiscence to occur. The incision size required for these multiport systems eliminates this advantage. The flexibility of the multiport system eliminated the problem of instrument interference that is inherent with the single port systems. The number of skin incisions is a cosmetic issue in the human field but is not a consideration in our patients. So to me the disadvantages of the single port systems far outweigh the advantages. Even with this bias I do not think that it affects my interpretation of this research.
I am really sorry that I cannot approve this publication as it is presented. I truly wish that I could. Does this need to be published anyway even with its deficiencies? I do not know. The message is important but do the deficiencies make the information that is presented inaccurate?
I have been thinking about ways to rewrite this to provide a more accurate representation of the information. I am having trouble doing this. Possibly retitling this as a pilot study with additional studies to follow with a different design using real instruments, measurements within the abdomen with an appropriate size and number of instruments would be an option. Determining the range of instrument motion needed to perform common minimally invasive surgical procedures would also be an important step.
Author Response
Dear reviewer,
thank you for reviewing our manuscript and all the efforts to improve it.
We are aware that the experimental setup comes with a couple of limitation. We added a section to the discussion to provide the reader a better understanding of our experimental setup. We also added the information why we placed the setup upside down in the Material and Method section to make it more easy to understand the experimental setup. (see line259 and 491ff)
Most of your concerns regarding the experimental setup comes with the motion capturing system. Since all the cameras were ceiling mounted, we decided to run the experiment upside down. The wooden dummy scopes were necessary to minimize refections form the scope or instrument tips influencing the measurements from the motion capture system.
As you mentioned, another limitation is the choice of endoscope size. Unfortunately, it was not possible to measure and calculate the movements of two instruments and an endoscope at the same time. since usually 2 Instruments are used in most procedures, we decided to use an 10mm dummy mimicking a 10mm endoscope with an incorporated working channel.
The asked for a GelPoint System unfortunately the system was not available when measurements where scheduled. Therefore, this system was not included in the study. Of course, the study would be more valuable the more ports are compared. We apologize for this circumstance.
We changed the term “acrylic glass box” to “clear acrylic box” as recommended.
As you mentioned, the main advantage of SILS for humans, the cosmetic benefits, are probably not an issue in veterinary medicine. Furthermore, small incisions to prevent dehiscence might be an advantage for a multiport approach. However, at least in our clinic, owners prefer one incision instead of multiple. We are also aware that owner’s expectations as treatment guidance might be another discussion.
As recommended, we changed the title to pilot study. We are planning to continue our research in this area and planning to investigate the maneuverability of different SPASs in standard training procedures such as the PEG transfer.
We are aware that there are specifically designed instruments when using SPASs which are shown to provide a higher maneuverability. We will include these instruments in future study focusing on the maneuverability while standard training procedures.
We hope that we improve the study according to your ideas and looking forward to your thought.
Best regards
Reviewer 2 Report
I would like to congratulate the Authors on the excellent manuscript.
Author Response
Dear Reviewer,
thank you for reviewing our manuscript. We hope you enjoyed reading it.
We used a professional English editing service before submitting the manuscript and apologize that they did not met do quality expected. We will revise the the manuscript and will enclose the editing certificate.
Best regards
Reviewer 3 Report
Dear authors,
the experimental setup is good in principle, but unfortunately not realistic and not sufficient to demonstrate the advantages or disadvantages of the different ports. Unfortunately, the tests were performed with only one endoscope-dummy and one instrument-dummy? This does not reflect a realistic setup and is therefore of limited value.
In addition, performing a simple exercise such as peg transfer and precision cutting (www.flsprogram.org) with a real endoscope and real instruments would be of interest. I would recommend using a 5mm/45° bariatric endoscope and 5mm straight instruments. The bariatric scope is longer than regular scope and this reduces external clashing of instruments. The tasks could be performed without time measurement, as this depends on the surgeon's skill level, but with determination of the grade of “clashing of the instruments”. This can certainly be mapped well with the method described here.
This would be of interest since a recent survey among pediatric surgeons revealed that “clashing of instruments’’ was perceived as the biggest problem of uniportal laparoscopic surgery (79% of the respondents) but that at the same time, only 12% of the surgeons used techniques that can limit this, such as long bariatric scopes. Additionally, commercially available ports were used by 51%; alternative devices such as home-made ports (e.g., ‘‘glove-port’’) were applied by 24% (Zimmermann P 2020).
I would appreciate a more detailed explaination why only two dummys (endoscope and instruments) have been used.
Perhaps there will be another study by your group with an experimental setup with more construct and content validity.
Ref.: Zimmermann P, Martynov I, Perger L, Scholz S, Lacher M. 20 Years of Single-Incision-Pediatric-Endoscopic-Surgery: A Survey on Opinion and Experience Among International Pediatric Endosurgery Group Members. J Laparoendosc Adv Surg Tech A. 2021 Mar;31(3):348-354. doi: 10.1089/lap.2020.0797. Epub 2020 Dec 31. PMID: 33395367.
Author Response
Dear Reviewer,
thank you for reviewing our manuscript. We hope you enjoyed reading it.
We are aware that there are some limitations in the experimental setup. The use of dummy endoscopes was given by the use of the motion capture system. The metal, and therefore reflecting, surface of the endosopces and the metal tips of instruments would have influenced the measurement. The ceiling mounted motion capture system was also the reason why we put the box upside down during measurement allowing an unhindered camera view. We added this limitations to the material and methods as well as to the discussion to make this more clear for the reader.
We are also aware, that a 10mm endoscope, even when including a 6mm working channel, are used seldom in veterinary medicine. Since it is impossible to measure the movements of two instruments with an 5mm endoscope, we decided that a 10mm endoscope (with working channel) and a 5mm instrument would be the most realistic situation we can transfer to our setting.
However, we focused more on this limitations in our revised discussion to explain the selection of the experimental more clearly since it do not reflect the most common situation.
We are planning to proceed with more studies in this topic focusing on a more realistic setup, including tasks like peg transfer in a second step. To underline this indent we also include pilot study in the title following the recommendation of one reviewers. Besides all limitation, we think that our study adds information to the current knowledge.
We used a professional editing service for English language and style. We will review the manuscript to improve English language and style. A certificate of the editing service can be enclose for your reference.
Best regards
Georg Haider
Reviewer 4 Report
NO COMMENTS
Author Response
Dear Reviewer,
thank you for reviewing our manuscript. We hope you enjoyed reading it.
We used a professional English editing service before submitting the manuscript. We will revise the manuscript and enclose a certificate the editing process.
Best regards
Georg Haider
Round 2
Reviewer 1 Report
I commend the authors in their efforts to address my concerns and that they are planning on doing additional studies.
I do still question the failure to use a GelPoint System.
I do not know if this was because none were available or none were donated for use in the study.